# Aronia Berry Supplementation Mitigates Inflammation in T Cell Transfer-Induced Colitis by Decreasing Oxidative Stress

**DOI:** 10.3390/nu11061316

**Published:** 2019-06-12

**Authors:** Ruisong Pei, Jiyuan Liu, Derek A. Martin, Jonathan C. Valdez, Justin Jeffery, Gregory A. Barrett-Wilt, Zhenhua Liu, Bradley W. Bolling

**Affiliations:** 1Department of Food Science, University of Wisconsin-Madison, 1605 Linden Dr., Madison, WI 53706, USA; ruisong.pei@wisc.edu (R.P.); jliu678@wisc.edu (J.L.); mrtn.drk@gmail.com (D.A.M.); jcvaldez@wisc.edu (J.C.V.); 2China Agricultural University, College of Food Science and Nutritional Engineering, No.17 Qinghua Dong Lu, Beijing 100083, China; 3Wisconsin Institutes for Medical Research, University of Wisconsin-Madison, 1111 Highland Ave., Madison, WI 53706, USA; jjjeffery@wisc.edu; 4Mass Spectrometry Facility, Biotechnology Center, University of Wisconsin-Madison, 425 Henry Mall, Madison, WI 53706, USA; barrettwilt@wisc.edu; 5University of Massachusetts, Amherst, School of Public Health and Health Sciences, 100 Holdsworth Way, Amherst, MA 01003, USA; zliu@nutrition.umass.edu

**Keywords:** aronia berry, adoptive transfer colitis, inflammatory bowel disease, oxidative stress

## Abstract

Oxidative stress is involved in the pathogenesis and progression of inflammatory bowel disease. Consumption of aronia berry inhibits T cell transfer colitis, but the antioxidant mechanisms pertinent to immune function are unclear. We hypothesized that aronia berry consumption could inhibit inflammation by modulating the antioxidant function of immunocytes and gastrointestinal tissues. Colitis was induced in recombinase activating gene-1 deficient (*Rag1*^-/-^) mice injected with syngeneic CD4^+^CD62L^+^ naïve T cells. Concurrent with transfer, mice consumed either 4.5% *w*/*w* aronia berry-supplemented or a control diet for five weeks. Aronia berry inhibited intestinal inflammation evidenced by lower colon weight/length ratios, 2-deoxy-2-[^18^F]fluoro-d-glucose (FDG) uptake, mRNA expressions of tumor necrosis factor alpha (TNF-α), and interferon gamma (IFN-γ) in the colon. Aronia berry also suppressed systemic inflammation evidenced by lower FDG uptake in the spleen, liver, and lung. Colitis induced increased colon malondialdehyde (MDA), decreased colon glutathione peroxidase (GPx) activity, reduced glutathione (rGSH) level, and suppressed expression of antioxidant enzymes in the colon and mesenteric lymph node (MLN). Aronia berry upregulated expression of antioxidant enzymes, prevented colitis-associated depletion of rGSH, and maintained GPx activity. Moreover, aronia berry modulated mitochondria-specific antioxidant activity and decreased splenic mitochondrial H_2_O_2_ production in colitic mice. Thus, aronia berry consumption inhibits oxidative stress in the colon during T cell transfer colitis because of its multifaceted antioxidant function in both the cytosol and mitochondria of immunocytes.

## 1. Introduction

Inflammatory bowel diseases (IBD) affect more than 0.3% of the population in North America, Oceania and many European countries, and the incidence is rising rapidly in newly industrialized countries in Africa, Asia and South America [1]. The primary therapeutics for IBD target inflammatory cascades and enteric microbiota [2,3]. For example, anti-tumor necrosis factor (TNF) antibody is widely used for downregulating immune response [4]. Antibiotics might selectively decrease pathogenic bacterial species or globally reduce bacterial population, depending on the spectrum of activity [5]. However, it has been documented that antibiotics might negatively affect the environmental conditions of gut microbiota and cause antimicrobial resistance [6]. Consumption of probiotics and prebiotics may increase beneficial commensal microbes such as *Lactobacillus* and *Bifidobacterium* species in the gut, but their treatment efficacy is not well established for IBD [5,7]. Moreover, prolonged treatment with these therapies can lead to high relapse rates, drug resistance and various adverse effects such as gastrointestinal problems, anemia, carcinogenesis, hepatotoxicity, nephrotoxicity and hypersensitivity reactions [6]. 

Recently, redox imbalance has been proposed as one potential etiologic factor for IBD [6]. Patients with IBD have decreased antioxidant capacity and increased biomarkers for oxidative stress in the blood relative to healthy individuals [8,9]. Oxidative stress is involved in the pathogenesis and progression of IBD via several mechanisms: reactive oxygen species (ROS) activate the nuclear factor kappa B (NF-κB) pathway, resulting in loss of the epithelial barrier via induction of metalloproteinases and myosin light-chain kinase [10]; lipid oxidation products such as 4-hydroxynonenal and oxysterols induce apoptosis of cells in the epithelial barrier [11,12]; moreover, ROS dose-dependently disrupt the actin cytoskeleton and monolayer barrier function by increasing actin oxidation [13].

Considering the limitations of current therapies and the important role of oxidative stress in pathogenesis and progression of IBD, various antioxidant-rich foods such as orange juice, blueberry extract, bergamot juice and grape juice have been examined for their beneficial effects on IBD [14,15,16,17]. Aronia is a genus of shrubs containing three major cultivars namely *Aronia melanocarpa*, *Aronia arbutifolia*, and *Aronia prunifolia*, while the predominant cultivar in the United States is *Aronia mitschurinii* ‘Viking’ which is likely hybridized from *A. melanocarpa* [18]. The aronia berry is rich in anthocyanins, flavonoids, hydroxycinnamic acids, and proanthocyanidins [18]. Consumption of aronia extract increased serum concentration of superoxide dismutase (SOD), glutathione peroxidase (GPx), and catalase in adults with metabolic syndrome relative to an untreated control group [19]. Aronia extract supplementation increased plasma paraoxonase and catalase activities, and hepatic GPx activity in apolipoprotein E knockout mice [20]. Aronia extract also inhibited expressions of lipopolysaccharide (LPS)-induced cyclooxygenase 2 (COX-2) and inducible nitric-oxide synthase (iNOS) in vitro [21]. 

We previously demonstrated that aronia berry consumption at a nutritionally-relevant dose inhibited T cell transfer-induced colitis in mice by modulation of anti-inflammatory Th17 and Treg differentiation, colonic cytokines, and cecal microbiota diversity [22]. However, it is not clear how the antioxidant functions of aronia berry contribute to the protective mechanism in T cell transfer colitis. We hypothesized that aronia berry consumption inhibits colitis by modulating antioxidant function of immunocytes. Here, we present evidence that aronia berry consumption prevents depletion of antioxidant enzymes and reduces oxidative stress in the immunocytes and tissues of mice with colitis, which contributes to the anti-inflammatory mechanism.

## 2. Material and Methods

### 2.1. Reagents

Acetonitrile (ACN), butylated hydroxytoluene (BHT), ethylenediaminetetraacetic acid (EDTA), ethanol, formic acid (FA), phosphate buffered saline (PBS), potassium chloride (KCl), sodium acetate, and sodium hydroxide (NaOH) were purchased from Fisher Scientific (Pittsburgh, PA, USA). 1,1,3,3-tetramethoxypropane (TMP), 2-thiobarbituric acid (TBA), 2-vinylpyridine, metaphosphoric acid, and trichloroacetic acid (TCA) were purchased from Sigma-Aldrich (St. Louis, MO, USA). (3-hydroxybenzyl) triphenylphosphonium bromide (MitoB), MitoPhenol (MitoP), d_15_-MitoB, and d_15_-MitoP were purchased from Cayman Chemical (Ann Arbor, MI, USA). DL-Dithiothreitol (DTT) was purchased from Dot Scientific (Burton, MI, USA). “Viking” aronia berry was provided by Bellbrook Berry Farm (Brooklyn, WI, USA).

### 2.2. Induction of Transfer Colitis and Experimental Design

Mice were obtained from the University of Wisconsin-Madison Research Animal Resource Center which maintained colonies from breeder pairs purchased from Jackson Laboratory (Bar Harbor, ME, USA). Mice were housed under controlled environment conditions with a 12-h light–dark cycle. All experiments were reviewed and approved by the Institutional Animal Care and Use Committee of the University of Wisconsin-Madison under protocol A005914-A02.

Colitis was induced by adoptive transfer as previously described [23]. Briefly, donor mice (C57BL/6J) were euthanized at 6–8 weeks old and spleens were harvested for isolation of CD4^+^CD62L^+^ naïve T cells using a naïve T cells isolation kit (Cat # 130-106-643, Miltenyi Biotec Inc., San Diego, CA, USA). Then, colitis was induced in C57BL/6J-background recombinase activating gene-1-deficient (*Rag1*^-/-^) mice by transferring 5 × 10^5^ purified CD4^+^CD62L^+^ naïve T cells intraperitoneally from gender and age-matched donor mice. After the transfer, the mice were randomly assigned to either the American Institute of Nutrition (AIN)-93M control diet as the colitic control group, or a modified AIN-93M diet supplemented with 4.5% lyophilized “Viking” aronia berry powder at the expense of corn starch (Envigo RMS, Inc., Indianapolis, IN, USA) as the aronia group. This dose is equivalent to a 70 kg adult consuming ~1 cup of fresh aronia berries per day [24]. The aronia-supplemented diet contained 847 ± 54 nmol phenolic acid, 1440 ± 250 nmol anthocyanin, and 233 ± 17 nmol proanthocyanidins (as (+)-catechin equivalents) per g of diet [22]. The dietary fiber content of aronia berry increased the total fiber content from 5% in the control diet to 6% in the aronia diet [25]. Another group of *Rag1*^-/-^ mice were reconstituted with splenic CD4^+^ T cells (containing a wide range of regulatory, helper, and effector T cells) isolated by a commercial kit (Cat# 130-104-454, Miltenyi Biotec Inc., San Diego, CA, USA), serving as the non-colitic control [26]. Food intake and body weights of mice were recorded for the duration of the experiment. Mice were euthanized at 5 weeks post transfer.

### 2.3. FDG Bio-Distribution

Tissue distribution of 2′-deoxy-2′-[^18^F]fluoro-d-glucose (FDG) was determined as a biomarker for tissue inflammation [27]. Briefly, mice were fasted 12 h prior to intravenous injection of approximately 1.85 MBq of FDG (Sofie Biosciences, Culver City, CA, USA) 1 h before sacrificing (*n* = 5–6 per group). Mice were then warmed after injection and induced with inhalation anesthetic gas using 2% isoflurane (Piramal Healthcare, Mumbai, India) mixed with 1 L/min of pure oxygen until cervical dislocation [28]. Colon, cecum, small intestine, spleen, liver, and lung were excised, wet-weighed using an XPE205 analytical scale (Mettler Toledo, Columbus, OH, USA) and then measured with a 2480 Wizard^2^ gamma counter (Perkin Elmer, Waltham, MA, USA). Blood and feces were flushed from the tissues, and surgical tools were rinsed between tissue resection to prevent cross contamination of FDG. Decay-corrected percent injection dose per gram of tissue (%ID/g_tissue_) was calculated for each sample.

### 2.4. GPx Activity

Colon samples were rinsed with ice-cold PBS and then homogenized in 5 mL of ice-cold PBS (with 5 mM EDTA and 1 mM DTT) per g tissue with a Fisher Scientific™ Bead Mill 4 homogenizer (Pittsburgh, PA, USA). The homogenate was then centrifuged at 10,000× *g* for 15 min at 4 °C. Supernatant aliquots were used to measure GPx activity by an assay kit (Cayman Chemical, Ann Arbor, MI) and to determine protein content by the Pierce™ BCA protein assay kit (Thermo Fisher Scientific, Waltham, MA, USA). GPx activity was expressed as nmol/min/µg protein.

### 2.5. Glutathione (GSH)

GSH was measured with the 5,5′-dithio-bis-2-(nitrobenzoic acid) assay using commercial reagents (Cayman Chemical, Ann Arbor, MI, USA) [29]. Supernatant was obtained similarly as described in Section 2.4, except that a PBS buffer at pH 7.0 with 1 mM EDTA was used for homogenization. Then, the supernatant was treated with an equal volume of metaphosphoric acid (10%, *m*/*v*) and centrifuged at 2000× *g* to remove interferences due to particulates and sulfhydryl groups on proteins. Half of the deproteinated sample was further treated with 2-vinylpyridine to derivatize reduced glutathione (rGSH) [30]. Then, the original and derivatized samples were used to determine total GSH content and glutathione disulfide (GSSG) only, respectively. GSH level was calculated as the percentage of rGSH in total GSH.

### 2.6. MitoB Treatment and Quantification

To determine the redox status in mitochondria, a ratiometric mass spectrometry probe namely MitoB was adopted; MitoB accumulates in mitochondria and reacts with H_2_O_2_ to form MitoP and the MitoP/MitoB ratio serves as an indicator of mitochondrial H_2_O_2_ level [31]. Briefly, mice received 25 nmol MitoB in sterile PBS by retro-orbital injection under anesthesia (*n* = 4–5 per group). Mice were then kept cages for 4 h to allow MitoB to respond to mitochondrial H_2_O_2_. After that, mice were euthanized, and the spleen and colon were harvested and cleaned with ice-cold PBS. For extraction, 40 mg of colon or spleen tissues were homogenized in 200 µL of 60% ACN/0.1% FA (*v*/*v*) in H_2_O with a Fisher Scientific™ Bead Mill 4 homogenizer (Pittsburgh, PA, USA). Then, the homogenates were spiked with 100 pmol d_15_-MitoB and 50 pmol d_15_-MitoP in 10 μL ethanol (as internal standard). Then, the mixture was centrifuged at 16,000× *g* for 10 min. The supernatant was filtered with a 0.22 µm polyvinylidene fluoride (PVDF) filter, dried with a Savant SpeedVac (Thermo Scientific, Waltham, MA, USA) and re-dissolved in 20% ACN/0.1% FA (*v*/*v*) in H_2_O for LC-MS/MS analysis.

Samples were resolved on an Inertsil Ph-3 (2.1mm × 150mm, 3 µm, GL Sciences, Rolling Hills Estates, CA, USA) using an Agilent 1100 series capillary HPLC system equipped with autosampler (held at 6 °C) and column compartment (held at 30 °C) (Agilent Technologies, Santa Clara, CA, USA). Solvents were A: 0.1% FA (*v*/*v*) in water, and B: 0.1% FA (*v*/*v*) in ACN running at 0.2 mL/min. The gradient started at 5% B for 2 min, then increased to 25% B over 3 min, increased to 75% B over 5 min, increased to 100% B over 5 min, held at 100% B for 5 min, decreased to 5% B over 5 min and held for 14 min. For mass spectrometry, a Thermo TSQ Quantum Discovery Max system (Thermo Fisher Scientific, Waltham, MA, USA) was operated in positive ion electrospray mode. Spray voltage was 4000 V using the HESI-II source. The vaporizer temperature was 400 °C. The sheath gas, ion sweep gas, and aux gas were set at 30, 0, and 1 (arbitrary units), respectively. The capillary temperature was 350 °C and capillary offset was 35 V. Selected reaction monitoring (SRM) data was collected in centroid mode with a scan width of 0.2 m/z and a scan time of 0.3 s. Collision energy was 50 V for all analytes. Collision gas pressure was 1.5 mT. For MitoP, Q1 was 369.22 and Q3 was 183.0. For d_15_-mitoP Q1 was 384.34 and Q3 was 191.1. For MitoB Q1 was 397.22 and Q3 was 183.0. For d_15_-mitoB Q1 was 412.34 and Q3 was 191.1. A calibration curve of standards was constructed using area under the curve of analytes (0–50 µM for MitoB; 0–10 µM for MitoP) relative to internal standards to determine analyte concentrations. 

### 2.7. Colon Malondialdehyde (MDA)

MDA in the colon was extracted and measured by high-performance liquid chromatography-fluorescence detection (HPLC-FLD) as previously described with minor modifications [32]. Briefly, 50 mg of colon was homogenized in 0.5 mL ice-cold 0.15 mol/L KCl containing 0.01% BHT using a Fisher Scientific™ Bead Mill 4 homogenizer (Pittsburgh, PA, USA). Then, 50 µL homogenate was mixed with 400 µL of KCl, 40 µL of 0.2% BHT in ethanol and 200 µL of 1N NaOH, and then incubated at 60 °C for 30 min. Then, protein was precipitated by mixing with 2 mL of 5% TCA. After centrifugation at 1000× *g* for 10 min at 4 °C, 500 µL of supernatant was derivatized with 0.6% TBA at 95 °C for 30 min. Then, the sample was extracted with butanol, and injected onto a Dionex UltiMate 3000 HPLC equipped with an LPG-3400 quaternary pump, a WPS-3000 analytical autosampler, and an FLD-3100 fluorescence detector (Thermo Fisher Scientific, San Jose, CA, USA). Samples were resolved isocratically at 0.8 mL/min on a Discovery^®^ C18 column (250 × 4.6 mm, 5µm; Supelco, Bellefonte, PA, USA) using 40:60 methanol and 25 mM potassium phosphate buffer (pH 6.5). MDA was quantified against standards prepared in parallel from TMP and was normalized to colonic protein.

### 2.8. qPCR Analysis

RNA in the colon, spleen, and mesenteric lymph node (MLN) was extracted by TRIzol reagent and further purified with a RNeasy mini kit (Qiagen, Valencia, CA, USA). After subsequent cDNA synthesis, reverse transcription quantitative PCR (RT-qPCR) was performed using the iTaq™ Universal SYBR^®^ Green Supermix (Bio-Rad, Hercules, CA, USA), on a Bio-Rad CFX96 system (Bio-Rad, Hercules, CA, USA). A detailed description of the methodology and primer sequences are provided in the Appendix A.

### 2.9. Statistical Analysis

All results were expressed as means ± SEMs. Statistical analysis was conducted on SAS 9.4 software (Cary, NC, USA). The significance level was set at α = 0.05 for all tests. Body weight changes were analyzed by two-way repeated measures (RM) ANOVA (PROC MIXED) with time and treatment group as independent variables. Multiple comparisons were conducted among groups at different weeks with Tukey’s test. For colon weight/length, FDG uptakes, MitoP/MitoB ratio, MDA, GPx activity, and rGSH level, significance testing was done by ANOVA with Tukey’s test for multiple comparison (PROC GLM). The mRNA expressions of cytokines in *Rag1*^-/-^ mice was assessed for significance by the Mann–Whitney U-test (PROC NPAR1WAY). The mRNA expressions of antioxidant enzymes and glucose transporters in *Rag1*^-/-^ mice was assessed for significance by the Kruskal–Wallis test with Dunn’s test for multiple comparison (PROC NPAR1WAY).

## 3. Results

### 3.1. Aronia Berry Inhibited Inflammation after T Cell Adoptive Transfer-Induced Colitis

Transfer of CD4^+^CD62L^+^ naïve T cells induced weight loss after five weeks in mice fed the control diet, but aronia berry consumption significantly reduced the extent of colitic weight loss (Figure 1A). At this time point, mice in both groups had developed severe colitis as indicated by histopathological analysis (Appendix A, [22]). The food intake in non-colitic, colitic, and aronia groups did not differ at week 5 (2.90 ± 0.14 vs. 2.63 ± 0.06 vs. 2.77 ± 0.17 g/mouse/day in male mice, *p* > 0.05; 2.40 ± 0.10 vs. 2.46 ± 0.40 vs. 2.37 ± 0.10 g/mouse/day in female mice, *p* > 0.05), indicating the supplementation of aronia powder did not change the palatability of diets. Naïve T cell transfer increased the colon weight/length ratio by 31% of the non-colitic group, whereas the aronia group had a similar colon weight/length ratio to the non-colitic group (Figure 1B). Colitis also increased the extent of inflammation as measured by FDG uptake in tissues. Relative to the non-colitic control, the colitic group had increased FDG uptake in the colon by 77%, small intestine by 50%, and spleen by 53%. In contrast, the aronia group had similar or lower FDG uptake in the colon, cecum, small intestine, spleen, liver, and lung compared to the non-colitic group and colitic group, indicating inhibition of inflammation in the gastrointestinal tract and peripheral tissues (Figure 1C). 

### 3.2. Aronia Berry Downregulated mRNA Expression of Inflammatory Cytokines

Relative to the colitic control, the aronia group had 86% and 48% less mRNA expression of tumor necrosis factor alpha (*Tnf*) and interferon gamma (*Ifng*) in colon tissue (Figure 2A). The colonic interleukin (IL)-6 (*Il6*) mRNA expression was 65% lower in the aronia group relative to the colitic control but was not statistically significant (*p* = 0.07). In the MLN, IL-17A (*Il17a*) and IL-10 (*Il10*) were both increased in the aronia group relative to the colitic control (Figure 2B). In the spleen, *Tnf, Ifng, Il17a,* and *Il10* were not affected by diet (Figure 2C). Likewise, *Foxp3* and *Rorc*, transcription factors indicative of Treg and Th17, respectively, were not altered by in the colon, MLN, or spleen by the aronia diet. 

### 3.3. Aronia Berry Reduced Colitis-Associated Oxidative Damage and Prevented Depletion of Antioxidant Enzymes

In the spleen, aronia consumption inhibited mitochondrial H_2_O_2_ by 42% relative to the colitic control group (Figure 3A). In contrast, the colonic MitoP/MitoB ratio did not differ among the non-colitic control, control, and aronia diet groups (Figure 3B). Colitis induced colonic MDA by 151%, but aronia consumption prevented this increase (Figure 4A). Compared with the non-colitic control, the colitic control had 16% lower colonic GPx activity (Figure 4B) and 44% lower colonic rGSH (Figure 4C), indicating depletion of antioxidant defenses. In contrast, aronia feeding maintained the level of colonic rGSH and GPx activity, which implicated the importance of thiols in preventing colitis-associated oxidative stress.

### 3.4. Aronia Berry Prevents Colitis-Associated Downregulation of Endogenous Antioxidant Enzymes mRNA Expression

The expression of nuclear factor-erythroid 2-related factor-2 (Nrf2) and antioxidant enzymes in the colon, MLN, and spleen was further evaluated to explain how aronia consumption modulates antioxidant function in these tissues. Colitis reduced colonic *Sod2*, *Gpx1*, and *Prdx1* (peroxiredoxin) by 37%, 29%, and 58%, respectively, compared to the non-colitic control (Figure 5A). Aronia consumption normalized *Sod2* and *Gpx1* to the non-colitic control. In the MLN, colitis also reduced the expression of Nrf2 (*Nfe2l2*), γ-glutamylcysteine synthetase (*Gclc*), glutathione reductase (*Gsr*), *Gpx1*, and *Gpx2* by 27% to 45% relative to the non-colitic control (Figure 5B). Aronia consumption also normalized the expression of these genes to the healthy control in the MLN. In the spleen, *Nfe2l2*, *Gclc*, *Gsr*, *Sod2*, *Gpx1*, *Gpx2*, and *Prdx1* were not consistently affected by T cell transfer colitis (Figure 5C). However, aronia supplementation tended to increase the expression of *Sod2* by 91% (*p* = 0.07), *Gpx1* by 127% (*p* = 0.09), and *Prdx1* by 121% (*p* = 0.07). 

### 3.5. Aronia Berry Does Not Affect mRNA Expression of Glucose Transporters

Since the uptake of FDG could be limited by glucose transport, we evaluated *Glut1* (glucose transporter 1) in the colon, MLN, and spleen, and *Sglt1* (sodium glucose cotransporter) in the colon of mice. Relative to the non-colitic control, adoptive transfer colitis induced *Glut1* by ~10-fold in the colon and by ~50% in the MLN (Figure 6A). Aronia feeding did not affect *Glut1* in the colon, MLN, or spleen nor colonic *Sglt1* in *Rag1*^-/-^ mice that received CD4^+^CD62L^+^ naïve T cells (Figure 6B). 

### 3.6. Aronia Berry Reduces Oxidative Stress in Healthy Wild Type Mice without Affecting Inflammatory Status

To determine whether the association between anti-inflammatory and antioxidant function of aronia berry depended on different physiological states, we evaluated how aronia consumption affected inflammation and redox status in healthy wild type mice that were fed either the control or the aronia diet for five weeks. Aronia berry supplementation did not change the expressions of splenic inflammatory cytokines (*Il17a, Il6, Il10, Tnf, Ifng*) or Treg/Th17 transcription factors (*Foxp3*, *Rorc*), tended to decrease *Gsr* (*p* = 0.06) and *Sod2* (*p* = 0.07), and decreased *Glut1* by 17% in healthy mice (Figure 7A–C). Since MLN tissue is an important site of T cell differentiation and aronia feeding modulated MLN expression of glutathione peroxidase during colitis, we evaluated mitochondrial H_2_O_2_ by MitoB. Aronia feeding decreased the mitochondrial H_2_O_2_ in MLN by 29% relative to the control diet (Figure 7D). 

## 4. Discussion

The T cell transfer model of colitis recapitulates the histopathological characteristics of human IBD [23]. In this rodent model of colitis, aronia consumption modulated oxidative stress in the colon, spleen, and MLN, tissues responsible for development of pro-inflammatory T cell populations. Aronia berry consumption inhibited colitic weight loss and reduced intestinal inflammation evidenced by lower colon weight/length ratios, reduced intestinal FDG uptake, and downregulation of proinflammatory cytokines in the colon. Colitis was associated with oxidative stress in the colon indicated by increased MDA production, decreased GPx activity, and rGSH level. Aronia berry consumption reduced the oxidative stress at least partly by preventing colitis-associated downregulation of *Nfe2l2* and antioxidant enzymes in colon and MLN.

We previously demonstrated that aronia feeding inhibited pro-inflammatory colonic cytokines and increased Th17 IL-10^+^ cells; however, colon histological markers were not affected [22]. Histopathological scores were based on a wide range of abnormalities such as degree of immunocytes infiltration, occurrence of crypt exudate, crypt loss, and/or effacement, presence of neutrophils and multinucleate giant cells, etc., which might have masked the ability to observe the anti-inflammatory effects of aronia berries by histopathological evaluation during severe colitis [22]. Moreover, the adoptive transfer of T cells not only induces colonic inflammation, but also causes inflammatory damage and injury in the small intestine, liver, and other organs [23]. Therefore, FDG bio-distribution was determined in various tissues to evaluate tissue-specific inflammatory activity. FDG is an analog of glucose which can be absorbed by cells through glucose transporters; unlike glucose, FDG cannot enter glycolysis and is trapped intracellularly as FDG-6-phosphate [33]. In the presence of inflammation, infiltration and activation of immune cells require increased glucose import primarily through GLUT1; therefore, FDG bio-distribution could serve as a non-invasive biomarker for tissue inflammation [27]. Since FDG bio-distribution primarily depends on the extent of immunocyte activation rather than other abnormalities in colitis, it may serve as a more effective technique to differentiate the anti-inflammatory effects of the dietary intervention at the late stage of colitis. In this study, the colitic control group had significantly higher FDG uptakes in the colon, cecum, and small intestine than non-colitic mice, indicating inflammation in the intestinal tract; by contrast, aronia feeding significantly reduced FDG uptake into tissues independent of modulation of glucose transporters. This is in accordance with our previous finding that adoptive transfer induced increased infiltration of immune cells in the colon, which were ameliorated by aronia supplementation [22]. Compared with colitic mice on control diet, the aronia group had lower expressions of proinflammatory cytokines in the colon. This might partly explain how aronia reduced adoptive transfer-induced production of proinflammatory cytokines in the colon [22]. Interestingly, the cytokine expression did not mirror the changes in transcription factors for Treg (FoxP3) and Th17 (ROR) cells differentiation in the colon as previously observed [22], although *Il17a* and *Il10* were increased in the MLN. The change in cytokines was more likely due to the extent of CD4^+^ cells infiltrating the colon [22]. Moreover, we observed that aronia feeding also suppressed FDG uptake in the spleen and lungs, demonstrating that the anti-inflammatory effect of aronia berry consumption during T cell transfer colitis extends to organs outside the intestinal tract. 

Loss of body weight and other symptoms of colitis (i.e., diarrhea or lose stool, reduced physical activity) result from systemic production of effector T cells (Th17 and Th1) after the T cell transfer and the interaction between effector T cells and gut microbiota [23,34]. Impaired absorption of nutrients during colitis can be explained by changes in enterocyte functionality in the small intestine [35]. The progression of colitis depends on mouse genetics and the housing environment [23]. We have consistently observed the onset of colitic weight loss beginning at four weeks after T cell transfer [22,36]. We previously showed that at seven weeks after adoptive transfer of T cells, mice developed severe colitis as measured by histopathology and colonic cytokine production [22]. However, the protective effect of aronia supplementation was diminished at this stage, suggesting a preventive rather than therapeutic effect.

These results reinforce the relationship of oxidative stress and colitis, evidenced by depletion of antioxidant enzymes and extent of oxidative damage. The rGSH and GPx activity in the colon was significantly lower in the colitic mice than in the non-colitic mice; by contrast, aronia feeding preserved rGSH which might also contribute to maintaining GPx activity. *Gcl*, the key enzyme involved in GSH synthesis, was downregulated in MLN of colitic control but sustained by aronia consumption. Similarly, decreased rGSH levels were found in both inflamed and non-inflamed mucosa in patients with IBD, which was attributed to reduced GCL activity [37]. Moreover, MDA was increased in colitic mice and reduced by aronia feeding. Similarly, MDA was elevated in the luminal epithelium of both inflamed Crohn’s disease (CD) and the inflamed ulcerative colitis (UC) mucosa [38]. 

During intestinal inflammation, the major source of ROS is from infiltrated immune cells, such as macrophages and neutrophils [10]. Aronia consumption inhibits myeloperoxidase (MPO) during T cell transfer, an indicator that neutrophil infiltration is suppressed during colitis [22]. Superoxide is produced in the cytosol by enzymes such as COX-2 and NADPH oxidase (NOX) and then converted into H_2_O_2_ by CuZn-SOD [6]. These enzymes are regulated by Nrf2, which modulates the cellular defense against oxidative insults within the intestinal epithelium and the lamina propria by induction of antioxidant enzymes [2]. Nrf2 deficient mice were more susceptible to dextran sulfate sodium (DSS)-induced colitis and developed much more severe symptoms than wild type mice, partly due to increased ROS production by COX-2 [39]. Nrf2 deficient mice also had greater LPS-induced pulmonary inflammation and greater NOX-dependent ROS production [40]. On the other hand, increased Nrf2 expression by enzymatically synthesized glycogen showed protective effects against DSS and 2,4,6-trinitrobenzenesulfonic acid (TNBS)-induced colitis in C57BL/6 mice [41]. In this study, Nrf2 transcription was suppressed in MLN of colitic mice and several downstream antioxidant enzymes were downregulated correspondingly. Aronia feeding was able to prevent the downregulation of Nrf2 in the MLN and maintain the transcription of antioxidant enzymes. Similarly, in a DSS-induced colitic mice model, peracetylated (-)-epigallocatechin-3-gallate abated colitis, partly by activation and maintenance of Nrf2 and the downstream antioxidant enzymes [42]. Interestingly, the mRNA level of Nrf2 in the colon was not affected by the T cell transfer, although the mRNA expression of SOD2 and *Gpx1* was lowered by colitis but maintained by aronia feeding. Aronia may have modulated Nrf2 protein concentration or nuclear translocation to affect the expression of downstream antioxidant enzymes in the colon.

Another important source of ROS is mitochondria, in which superoxide is produced by the electron transport chain during the respiratory process [6]. Mitochondrial ROS levels in mononuclear cells were significantly higher in patients with IBD [43]. Moreover, dinitrophenol-induced mitochondrial superoxide caused significant internalization and translocation of *Escherichia coli* across epithelia in biopsy specimens in patients with CD [44]. However, mitochondria ROS is underestimated due to lack of reliable quantification methods in vivo; a ratiometric mass spectrometry probe MitoB which reacts with H_2_O_2_ and accumulates in mitochondria facilitates the measurement of mitochondrial H_2_O_2_ production in vivo [31]. In this study, mitochondrial H_2_O_2_ level, indicated by MitoP/MitoB ratios did not differ in colons of the colitic and aronia groups, but was significantly lowered by aronia feeding in the spleen corroborating reduced splenic FDG. The different effects of aronia supplementation on mitochondrial H_2_O_2_ production in the spleen and colon during colitis could be explained by differences in cell populations between these tissues. We cannot exclude the possibility that colonic epithelial cells or immunocytes have altered mitochondrial H_2_O_2_ production after aronia consumptions, since we evaluated MitoP/MitoB ratios in whole colon tissue. Indeed, anthocyanin and other berry polyphenols can have direct effects on epithelial cells, which include inhibition of NF-κB, inhibiting cytokine production, and improving barrier function [45]. Isolation of specific cell types from the colon and other tissues requires additional time and processing, which may alter transient MitoP/MitoB distribution in viable cells. Thus, other experimental approaches are needed to define the role of mitochondrial H_2_O_2_ in specific cell types during colitis.

These data are supported by the more robust modulation of antioxidant enzymes in the MLN relative to the colon in response to aronia supplementation. Similarly, a mitochondrial-targeted derivative of the antioxidant ubiquinone significantly ameliorated DSS-induced colitis by suppressing mitochondrial injury [43]. Another mitochondria-targeted antioxidant, (2-(2,2,6,6-Tetramethylpiperidin-1-oxyl-4-ylamino)-2-oxoethyl) triphenylphosphonium chloride inhibited the barrier defects and reduced the severity of DSS-induced colitis [44]. Interestingly, we also showed that aronia feeding decreased MLN mitochondrial H_2_O_2_ level in healthy wild-type mice. Because superoxide is produced by the electron transport chain during the respiratory process under physiological conditions, aronia berry reduces basal mitochondrial oxidative stress prior to the initiation of chronic intestinal inflammation.

Polyphenol-rich foods are protective in rodent models of colitis. In a DSS-induced colitis mice model, orally administered blueberry extract attenuated the development of colitis, which was associated with markedly attenuated colonic MPO activity, decreased MDA in the colon, increased serum levels of SOD and catalase [15]. In a dinitrobenzene sulfonic acid (DNBS)-induced colitis model, daily feeding of flavonoid-rich extract of orange juice decreased neutrophil infiltration, colonic TNF-α and IL-1β generation, and nuclear NF-κB translocation, which were associated with decreased nitrotyrosine and enhanced MnSOD expression [14]. These models are based on chemical erosion and cannot recapitulate many histopathological characteristics of human IBD; by contrast, the T cell adoptive transfer model induces intestinal inflammation that shares the same histopathological characteristics as human IBD [34]. Increasing evidence supports the anti-inflammatory effects of dietary fiber in IBD. For example, Hung and Suzuki examined the effects of different doses of dietary fiber supplementation (5% and 10%) in DSS-induced mice and found that 10% of dietary fiber reduced intestinal barrier defects and inflammation [46]. Similarly, supplementation of 10% of pectin soluble fiber attenuated clinical and inflammatory markers of acute and chronic DSS-induced colitis in mice [47]. In this study, the fiber content in the control and aronia diets were ~5%–6% (*w*/*w*), which was lower than the previously reported effective dose. Nevertheless, future work is warranted to examine the potential synergistic effects between fiber and polyphenols in aronia berries.

In conclusion, we demonstrated that the antioxidant effects of aronia are multifaceted in the T cell adoptive transfer model of colitis. Aronia feeding upregulated expression of antioxidant enzymes, prevented colitis-induced depletion of rGSH, maintained GPx activity, and also targeted mitochondria and reduced mitochondrial ROS production. These changes illustrate that antioxidant mechanisms of aronia consumption during T cell transfer colitis are multi-faceted and tissue specific. This study illustrates that the protective effects of aronia consumption during colitis are closely associated with antioxidant and mitochondrial peroxide levels in immunocytes.

## Figures and Tables

**Figure 1 nutrients-11-01316-f001:**
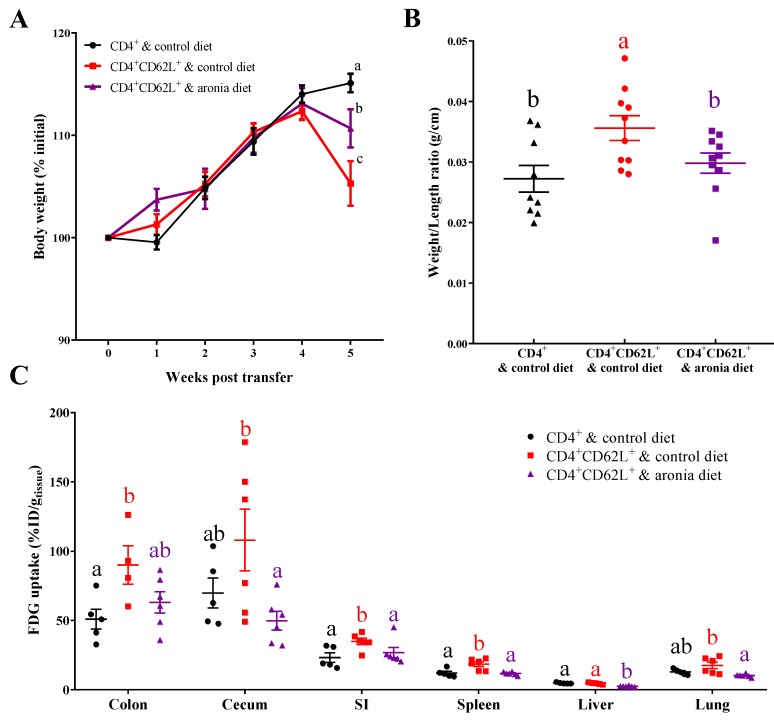
Consumption of 4.5% aronia berry-supplemented diet mitigates wasting induced by T cell transfer colitis in mice. Splenic CD4^+^CD62L^+^ cells from C57BL/6J mice were transferred to *Rag1*^-/-^ mice. Mice consumed the AIN-93M diet (colitic control) or aronia-supplemented diet for five weeks. A third group of *Rag1*^-/-^ mice received splenic CD4^+^ cells, consumed AIN-93M diet, and served as the non-colitic control. (**A**) Body weight changes after T cell transfer, as mean percentage of initial body weight at transfer (*n* = 9–10/group). Data bearing different letters indicate significant within-week differences (*p* < 0.05). (**B**) Colon weight/length ratio (*n* = 9–10/group). (**C**) Tissue 2-deoxy-2-[^18^F]fluoro-d-glucose (FDG) uptakes at 1 h after intravenous injection of approximately 1.85 MBq of FDG (*n* = 4–6/group). Groups bearing different letters indicate significant differences between treatments (*p* < 0.05). Data are means ± SEMs.

**Figure 2 nutrients-11-01316-f002:**
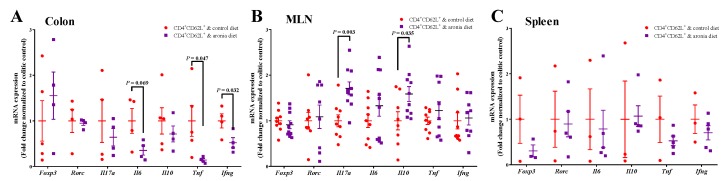
Aronia consumption downregulates mRNA expression of pro-inflammatory cytokines and up-regulates anti-inflammatory cytokines. Splenic CD4^+^CD62L^+^ cells from C57BL/6J mice were transferred to *Rag1*^-/-^ mice. Mice consumed the AIN-93M diet (colitic control) or aronia-supplemented diet for five weeks. A third group of *Rag1*^-/-^ mice received splenic CD4^+^ cells, consumed AIN-93M diet and served as the non-colitic control. The mRNA expressions of transcription factors of *Foxp3* (forkhead box P3) and *Rorc* (RAR-related orphan receptor gamma), and cytokines of *Il17a* (interleukin 17A), *Il22* (interleukin 22), *Il6* (interleukin 6), *Il10* (interleukin 10), *Ifng* (interferon-gamma), and *Tnf* (tumor necrosis factors) in the (**A**) colon (*n* = 4–5/group), (**B**) mesenteric lymph node (MLN) (*n* = 8–10/group), and (**C**) spleen (*n* = 3–5/group). The mRNA expressions were normalized to *Eef2* (eukaryotic translation elongation factor 2) and *Rplp0* (ribosomal protein large P0) genes. Data are means ± SEMs.

**Figure 3 nutrients-11-01316-f003:**
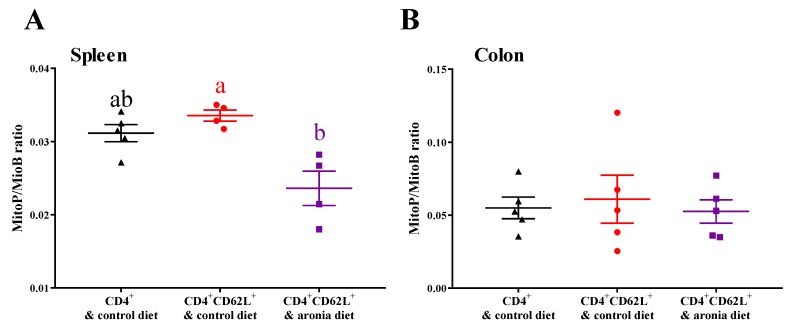
Aronia consumption reduces mitochondria H_2_O_2_. Splenic CD4^+^CD62L^+^ cells from C57BL/6J mice were transferred to *Rag1*^-/-^ mice. Mice consumed the AIN-93M diet (colitic control) or aronia-supplemented diet for five weeks. A third group of *Rag1*^-/-^ mice received splenic CD4^+^ cells, consumed AIN-93M diet, and served as non-colitic control. MitoP/MitoB ratio in (**A**) spleen and (**B**) colon (*n* = 4–5/group). Groups bearing different letters indicate significant differences between treatments (*p* < 0.05). Data are means ± SEMs.

**Figure 4 nutrients-11-01316-f004:**
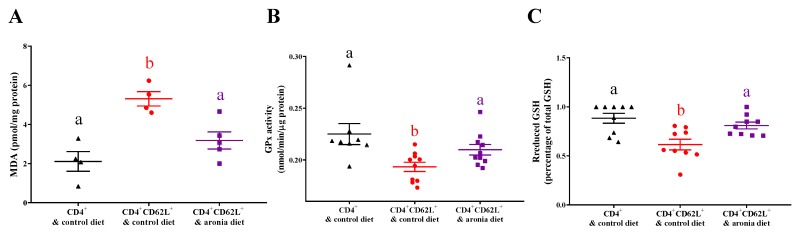
Aronia consumption reduces malondialdehyde (MDA) and prevents depletion of antioxidant enzymes in colon. Splenic CD4^+^CD62L^+^ cells from C57BL/6J mice were transferred to *Rag1*^-/-^ mice. Mice consumed the AIN-93M diet (colitic control) or aronia-supplemented diet for five weeks. A third group of *Rag1*^-/-^ mice received splenic CD4^+^ cells, consumed AIN-93M diet, and served as the non-colitic control. (**A**) Colon MDA (*n* = 4–5/group). (**B**) Colon glutathione peroxidase (GPx) activity (*n* = 8–10/group). (**C**) Colon reduced glutathione (rGSH) (*n* = 8–10/group). Groups bearing different letters indicate significant differences between treatments (*p* < 0.05). Data are means ± SEMs.

**Figure 5 nutrients-11-01316-f005:**
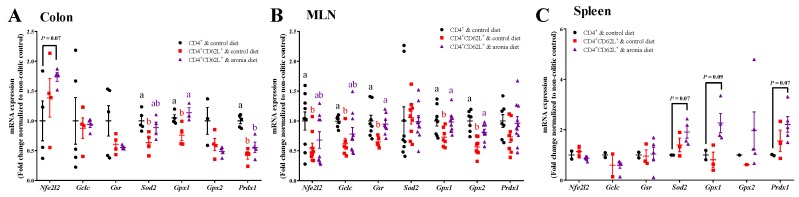
Aronia consumption maintains mRNA expression of endogenous antioxidant enzymes. Splenic CD4^+^CD62L^+^ cells from C57BL/6J mice were transferred to *Rag1*^-/-^ mice. Mice consumed the AIN-93M diet (colitic control) or aronia-supplemented diet for five weeks. A third group of *Rag1*^-/-^ mice received splenic CD4^+^ cells, consumed AIN-93M diet, and served as the non-colitic control. The mRNA expressions of transcription factor of Nfe*2l2* (erythroid-derived 2-like 2) and the downstream antioxidant enzymes of *Gclc* (glutamate-cysteine ligase-catalytic subunit), *Gsr* (glutathione reductase), *Sod2* (superoxide dismutase 2), *Gpx1* (glutathione peroxidase 1), *Gpx2* (glutathione peroxidase 2), and *Prdx1* (peroxiredoxin 1) in the (**A**) colon (*n* = 4–5/group), (**B**) mesenteric lymph node (MLN) (*n* = 9–10/group), and (**C**) spleen (*n* = 3–5/group). Gene expression was normalized to *Eef2* (eukaryotic translation elongation factor 2) and *Rplp0* (ribosomal protein large P0) genes. Bars bearing different letters indicate significant differences between treatments (*p* < 0.05). Data are means ± SEMs.

**Figure 6 nutrients-11-01316-f006:**
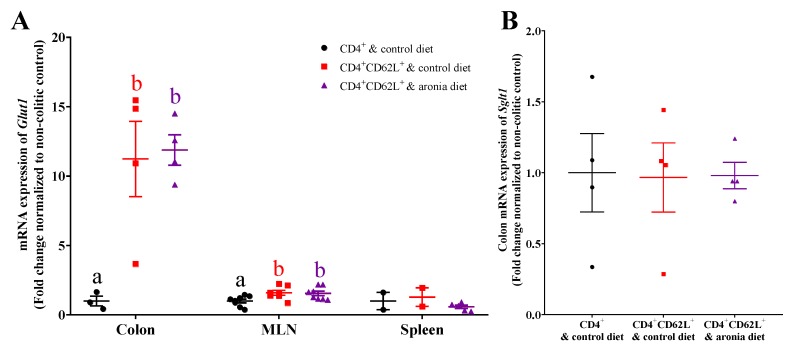
Aronia berry does not affect the expression of glucose transporters in colitic mice. Splenic CD4^+^CD62L^+^ cells from C57BL/6J mice were transferred to *Rag1*^-/-^ mice. Mice consumed the AIN-93M diet (colitic control) or aronia-supplemented diet for five weeks. A third group of *Rag1*^-/-^ mice received splenic CD4^+^ cells, consumed AIN-93M diet, and served as the non-colitic control. (**A**) *Glut1* (glucose transporter 1) in the colon (*n* = 4–5/group), mesenteric lymph node (MLN) (*n* = 9–10/group), and spleen (*n* = 3–5/group). (**B**) *Sglt1* (sodium glucose cotransporter 1) expression in colon (*n* = 4–5/group). The mRNA expressions in *Rag1*^-/-^ mice were normalized to *Eef2* (eukaryotic translation elongation factor 2) and *Rplp0* (ribosomal protein large P0) genes. Bars bearing different letters indicate significant differences between treatments (*p* < 0.05).

**Figure 7 nutrients-11-01316-f007:**
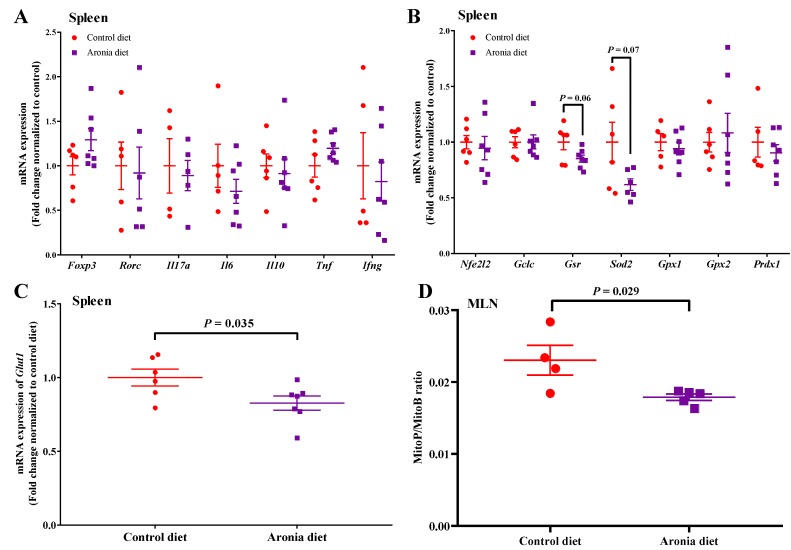
Aronia consumption decreases mitochondria H_2_O_2_ in MLN but does not affect the expression of antioxidant enzymes and cytokines in the spleen. C57BL/6J mice consumed the AIN-93M diet or aronia-supplemented diet for five weeks. (**A**) The mRNA expressions of transcription factors of *Foxp3* (forkhead box P3) and *Rorc* (RAR-related orphan receptor gamma), and cytokines of *Il17a* (interleukin 17A), *Il6* (interleukin 6), *Il10* (interleukin 10), *Ifng* (interferon-γ), and *Tnf* (tumor necrosis factors) in the spleen (*n* = 5–7/group). (**B**) The mRNA expressions of transcription factor of Nfe*2l2* (erythroid-derived 2-like 2) and the downstream antioxidant enzymes of *Gclc* (glutamate-cysteine ligase-catalytic subunit), *Gsr* (glutathione reductase), *Sod2* (superoxide dismutase 2), *Gpx1* (glutathione peroxidase 1), *Gpx2* (glutathione peroxidase 2), and *Prdx1* (peroxiredoxin 1) in the spleen (*n* = 6–7/group). (**C**) The mRNA expressions of *Glut1* (glucose transporter 1) in the spleen (*n* = 6–7/group). (**D**) MitoP/MitoB ratio in mesenteric lymph node (MLN) (*n* = 4–5/group). The mRNA expressions (normalized to *Eef2* (eukaryotic translation elongation factor 2) and *Rplp0* (ribosomal protein large P0)). Data are means ± SEMs.

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
