# Peer review of "Aronia Berry Supplementation Mitigates Inflammation in T Cell Transfer-Induced Colitis by Decreasing Oxidative Stress"

_nutrients, 2019, doi:10.3390/nu11061316_

Reviewer 1 Report

The study by the authors Pei et al. claim that the oxidative stress is a central mechanism involved in the pathogenesis and progression of inflammatory bowel disease. In their study, the authors compared colitis development (model of transfer colitis) in an aronia supplemented vs control diet.

They claim that food supplementation with aronia berries inhibit T cell transfer colitis as stated in the abstract, which is overinterpreted and not reflected by the data. 

Major concerns:

Title is wrong! There is no inhibition of colitis - the most you could state improved colitis! 

Fig1a: kinetics of body weight should be evaluated even beyond week 5 (6 and 7). In order to evaluate the effect of ariona berries, a detailed histological analysis of the colon is mandatory.

The phenotype of the transferred CD4+ CD62L+ T cells should also be investigated at different time points after transfer. Does aronia supplementation influence the function and phenotype of these effector T cells?

Significant changes between groups should not be indicated by bars and asterics as shown in Fig.2 and not by letters! Please show individual data points in your graphs, which allows better evaluation of the distribution of indivvidual points. 

Fig.3 and 5: It is difficult to understand and counter-intuitive why oxidative stress and expression of antioxidant enzymes are more affected in the spleen than in the colon?

In general: There are several mistakes concering the labeling of the axis, including T cell populations (CD62L+) and diet.

 Author Response

RESPONSE TO PEER REVIEW

We sincerely appreciate the constructive comments provided by the peer reviewers and editor. The following is our reply and explanation of changes made to the manuscript considering these comments. Changes to the text are marked in red.

Comment #1: The study by the authors Pei et al. claim that the oxidative stress is a central mechanism involved in the pathogenesis and progression of inflammatory bowel disease. In their study, the authors compared colitis development (model of transfer colitis) in an aronia supplemented vs control diet. They claim that food supplementation with aronia berries inhibit T cell transfer colitis as stated in the abstract, which is overinterpreted and not reflected by the data. 

Major concerns:

Title is wrong! There is no inhibition of colitis - the most you could state improved colitis! 

Response #1: We have revised the title to be, “Aronia berry supplementation inhibits inflammation in T cell transfer-induced colitis by decreasing oxidative stress”.

Comment #2: Fig1a: kinetics of body weight should be evaluated even beyond week 5 (6 and 7). In order to evaluate the effect of aronia berries, a detailed histological analysis of the colon is mandatory.

Response #2:  We previously evaluated body weight loss until week 7 using the same experimental design [1]. In this study, we chose to focus on week 5 for analysis of oxidative stress because the cell differences were more apparent at this time point [1]. We also conducted detailed histological analysis at multiple time points after adoptive transfer as previously reported [1]. Importantly, there were no differences in histological scoring of colitis at weeks 3, 5, or 7, despite inhibition of cytokines, weight loss, and colonic edema [1]. We have compiled these data in the Supplementary Information for reference. In consultation with our collaborating pathologist, we concluded there was a narrow range of colitis severity where we could differentiate the aronia treatment via histopathology. Aronia consumption did inhibit histopathological colitis scoring at week 3, when the naïve T cells were from IL-10, knock-out mice, because the onset of colitis was more rapid compared to transfer of T cells from wild type mice. We did not repeat histopathological analysis at week 5 for the present study because we concluded it was unnecessary because  1) it was likely we would achieve the same outcome in duplicating the analysis as in our prior report [1]; 2) the analysis required the whole colon due to the spontaneous nature of the colitis so would induce unnecessary pain and suffering of mice involved in these experiments; 3) FDG analysis would allow for a more precise analysis of inflammation that could better account for the extent of active inflammation  [2,3][4]. Since the mechanism of FDG determines active inflammation rather than histopathogical abnormalities in colitis, we pursued this analysis to determine inflammation in the colon and other tissues.

We have added the following discussion at LN 245 and 327 to clarify:

LN 245: At this time point, mice in both groups had developed severe colitis as indicated by histopathological analysis (Supplementary Fig. S1, [22]).

LN 327: … however, colon histological markers were not affected [22], excepted in mice transferred with naïve T cells from IL-10-/- mice (which yielded a more aggressive colitis model) at the onset of colitis [22]. Histopathological scores were based on a wide range of abnormalities such as degree of immunocytes infiltration, occurrence of crypt exudate, crypt loss, and/or effacement, presence of neutrophils and mutinucleate giant cells, etc., which might have masked the ability to observe the anti-inflammatory effects of aronia berries by histopathological evaluation during severe colitis [22].

Comment #3: The phenotype of the transferred CD4+ CD62L+ T cells should also be investigated at different time points after transfer. Does aronia supplementation influence the function and phenotype of these effector T cells?

Response #3:  We appreciate this suggestion and have previously reported this data. The transferred CD4+ CD62L+ (naïve) T cells differentiate to effector and helper cell phenotypes in the adoptive transfer model. Our prior study extensively evaluated T cell populations in the colon, spleen, and MLN using the same model, dietary intervention, and timespan. Thus, reporting this data here would be extensively redundant with our previous paper. Aronia supplementation indeed influenced the differentiation of these naïve T cells. We have previously shown that aronia supplementation increased the Treg and regulatory Th17 subpopulations (IL-17A+IL-10+ and IL-17A+IL-22+) in lamina propria and spleen [1].  These results are discussed in the Introduction (LN 94-97) and Discussion (LN 352-355).

Comment #4: Significant changes between groups should not be indicated by bars and asterics as shown in Fig.2 and not by letters! Please show individual data points in your graphs, which allows better evaluation of the distribution of individual points. 

Response #4:  We have revised the figures to show individual data points as suggested.

Comment #5: Fig.3 and 5: It is difficult to understand and counter-intuitive why oxidative stress and expression of antioxidant enzymes are more affected in the spleen than in the colon?

Response #5: We would like to clarify that oxidative stress was induced in the colon (Fig. 4) and aronia consumption dampened oxidative stress. In the colon, Gpx1 was increased (Fig. 5). We attribute the more robust differences in antioxidant enzyme gene expression in the spleen and MLN to the differences in cell types and immune compartmentalization. The cell populations in spleen/MLN and colon are vastly different. The spleen is a secondary lymphoid organ that is mainly made of immunocytes while immunocytes contribute only a small fraction to colon cell population. Our data suggest that mitochondria in immunocytes might be a more important source of ROS in IBD. The role of immunocytes in redox signing during IBD is further supported by our data that the mRNA expression in MLN is more responsive to colitis and aronia supplementation than that in colon.

Figure 3 demonstrates mitochondrial ROS was dampened in the spleen, but not the colon. Because the redox modulation was more apparent in immunocyte-rich tissue (spleen/MLN) it is likely that any changes to mitochondrial immunocytes in the colon were insufficient to lead to changes in MitoP/MitoB ratio in whole colon tissue. We considered isolating lymphocytes from colon tissue to evaluate MitoP/MitoB ratios, but determined that the isolation and purification process was too would further induce oxidative stress and cell death, so we would not have confidence in the outcome.

Comment #6: In general: There are several mistakes concerning the labeling of the axis, including T cell populations (CD62L+) and diet.

Response #6:   We have reviewed and corrected the labels in the figures.

1.         Pei, R.; Martin, D.A.; Valdez, J.; Liu, J.; Kerby, R.; Rey, F.; Smyth, J.; Liu, Z.; Bolling, B.W. Dietary prevention of colitis by aronia berry is mediated through t cell il-10 and increased th17 and treg. Mol. Nutr. Food Res. 2018.

2.         Brewer, S.; McPherson, M.; Fujiwara, D.; Turovskaya, O.; Ziring, D.; Chen, L.; Takedatsu, H.; Targan, S.R.; Wei, B.; Braun, J. Molecular imaging of murine intestinal inflammation with 2-deoxy-2-[f-18]fluoro-d-glucose and positron emission tomography. Gastroenterology 2008, 135, 744-755.

3.         Bicik, I.; Bauerfeind, P.; Breitbach, T.; vonSchulthess, G.K.; Fried, M. Inflammatory bowel disease activity measured by positron-emission tomography. Lancet 1997, 350, 262-262.

4.         Heylen, M.; Deleye, S.; De Man, J.G.; Ruyssers, N.E.; Vermeulen, W.; Stroobants, S.; Pelckmans, P.A.; Moreels, T.G.; Staelens, S.; De Winter, B.Y. Colonoscopy and mu pet/ct are valid techniques to monitor inflammation in the adoptive transfer colitis model in mice. Inflamm. Bowel Dis. 2013, 19, 967-976.

Reviewer 2 Report

The present manuscript addresses how Aronia berry consumption inhibits oxidative stress in an induced model of colitis by exerting an antioxidant function in both the cytosol and mitochondria of immunocytes.

There are a few concern from my part.

Minor comments

*Please define Aronia berry

*TNF-α , infg….:Authors sometimes use greek simbols when they name some interleukines but other they don´t.

Please, revise.

*Line 47:  Inflammatory bowel diseases affect 1.3% of US adults: US:

Do authors have an estimation of IBD incidence worldwide?

*Line 48: the term microflora has been updated and it is currently more used microbiota

 *Please revise abbreviations along the text: some are not defined in the right place, other are not defined  at all and other are sometimes  used in the abbreviated form but other they do not  ( i.e. …were euthanized at 6-8 wk and   ….were euthanized at 5 weeks, hours and h, etc.)

*Line 119:  until cervical dislocation….. liver, and,…

Last comma should to be removed.

*Line 206…. diet, but Aronia berry consumption significantly reduced the extent of colitic wasting (Fig. 1A).

Figure 1 A shows changes in weight. It is not clear for this reviewer why authors mention the extent of colitic wasting

*Line 303: … significance by Kruskal-Wallis test with Dunn’s test for multiple comparison (PROC NPAR1WAY…

Statistical analysis has already been described in material and methods section

 Mayor comments

 *Line 50:  Antibiotics, probiotics, and prebiotics are also commonly used for treating IBD via altering the composition of gut microbiota ….. therapies can lead to high relapse rates, drug resistance, and various adverse effects such as gastrointestinal problems, anemia, carcinogenesis, hepatotoxicity, nephrotoxicity and hypersensitivity reactions.

  Could authors discuss current controversies about the use antibiotics, probiotics, and prebiotics and how they use alter microbiota?

  *CD4+CD62L+ and CD4+ groups:

Which are the differences between these groups?

 *In addition to aronia powder, which are the differences between the diets of the different groups?

 *Do authors found any gender dimorphism in their results?

 *Do authors have any data about the effect of higher doses of Aronia berry?

  *Line 112:  Mice were euthanized at 5 weeks post transfer. However authors also say that (starting at line 392) “at 7 weeks after adoptive transfer of T cells, mice developed severe colitis as measured by histopathology and colonic cytokine production. However, the protective effect of aronia supplementation was diminished at this stage, suggesting a preventive rather than therapeutic approach.

 If animals were euthanized at 5 weeks it is not clear how authors discuss the effect of aronia at week 7.

 Author Response

RESPONSE TO PEER REVIEW

We sincerely appreciate the constructive comments provided by the peer reviewers and editor. The following is our reply and explanation of changes made to the manuscript considering these comments. Changes to the text are marked in red.

Comment #1: The present manuscript addresses how Aronia berry consumption inhibits oxidative stress in an induced model of colitis by exerting an antioxidant function in both the cytosol and mitochondria of immunocytes. There are a few concerns from my part.

Minor comments

*Please define Aronia berry

Response #1:  We have added the following description about aronia berry at LN 82-85:

Aronia is a genus of shrubs containing three major cultivars namely A. melanocarpa, A. arbutifolia, and A. prunifolia, while the predominant cultivar in the United States is A. mitschurinii ‘Viking’ which is likely hybridized from A. melanocarpa [17].

Comment #2: *TNF-α, infg….: Authors sometimes use Greek symbols when they name some interleukins but other they don´t.

Please, revise.

Response #2  We have revised the text for consistency. We followed gene naming conventions in some cases, which do not contain Greek lettering.

Comment #3: *Line 47:  Inflammatory bowel diseases affect 1.3% of US adults: US:

Do authors have an estimation of IBD incidence worldwide?

Response #3:  We have added the following information at LN 56-58:

Inflammatory bowel diseases (IBD) affect more than 0.3% in North America, Oceania, and many European countries, and the incidence is rising rapidly in newly industrialized countries in Africa, Asia, and South America [1].

Comment #4: *Line 48: the term microflora has been updated and it is currently more used microbiota

Response #4:  We have changed the term as suggested at LN 59.

 Comment #5: *Please revise abbreviations along the text: some are not defined in the right place, other are not defined at all and other are sometimes used in the abbreviated form but other they do not (i.e. …were euthanized at 6-8 wk and   …. were euthanized at 5 weeks, hours and h, etc.)

Response #5:  We have corrected the use of abbreviations as suggested.

Comment #6: *Line 119:  until cervical dislocation….. liver, and,…

Last comma should to be removed.

Response #6:  The comma has been removed.

Comment #13: *Line 206…. diet, but Aronia berry consumption significantly reduced the extent of colitic wasting (Fig. 1A). Figure 1 A shows changes in weight. It is not clear for this reviewer why authors mention the extent of colitic wasting

Response #7:  We have changed “wasting” this to “weight loss” for clarity.

Comment #14: *Line 303: … significance by Kruskal-Wallis test with Dunn’s test for multiple comparison (PROC NPAR1WAY…Statistical analysis has already been described in material and methods section.

Response #7:  We have removed the detailed description of statistical analysis from the figure legends as requested.

Comment #8: Major comments

*Line 50:  Antibiotics, probiotics, and prebiotics are also commonly used for treating IBD via altering the composition of gut microbiota …. therapies can lead to high relapse rates, drug resistance, and various adverse effects such as gastrointestinal problems, anemia, carcinogenesis, hepatotoxicity, nephrotoxicity and hypersensitivity reactions.

Could authors discuss current controversies about the use antibiotics, probiotics, and prebiotics and how they use alter microbiota?

Response #8:  We have added the following discussion at LN 61-67:

Antibiotics might selectively decrease pathogenic bacterial species or globally reduce bacterial population, depending on the spectrum of activity [5]. However, it has been well documented that antibiotics might negatively affect the environmental conditions of gut microbiota and cause antimicrobial resistance [6]. Consumption of probiotics and prebiotics may increase beneficial commensal microbes such as Lactobacillus and Bifidobacterium species in the gut, but their treatment efficacy is not well established for IBD [5,7].

Comment #9: *CD4+CD62L+ and CD4+ groups:

Which are the differences between these groups?

Response #9:  CD4+CD62L+ cells are naïve T cells (LN 123) that have not encountered by antigens and differentiate into effect T cells. CD4+ cells are a wide range of T helper cells including T regulatory cells which can resolve inflammation and prevent colitis. Therefore, CD4+ cells are injected as healthy control.

We added the following to clarify the latter at LN 134:

Another group of Rag1-/- mice were reconstituted with splenic CD4+ T cells (containing a wide range of regulatory, helper and effector T cells) isolated by a commercial kit (Cat# 130-104-454, Miltenyi Biotec Inc., San Diego, CA, USA), serving as the non-colitic control [25].

Comment #10: *In addition to aronia powder, which are the differences between the diets of the different groups?

Response #10:  We added 4.5% lyophilized “Viking” aronia berry powder at the expense of corn starch. So there is small difference in fiber content (5% vs. 6%), which is significantly lower than the previously reported effective dose, as we discussed at LN 437-444.

Comment #11: *Do authors found any gender dimorphism in their results?

Response #11: Statistical analysis indicated gender did not affect the study outcomes. Therefore, gender was not discussed in the results.

Comment #12: *Do authors have any data about the effect of higher doses of Aronia berry?

Response #12:  We chose the dosage of 4.5% because this dose is equivalent to a 70 kg adult consuming ~ 1 cup of fresh aronia berries per day. Therefore, this dose is more physiologically relevant and achievable by daily consumption.

Comment #13: *Line 112:  Mice were euthanized at 5 weeks post transfer. However, authors also say that (starting at line 392) “at 7 weeks after adoptive transfer of T cells, mice developed severe colitis as measured by histopathology and colonic cytokine production. However, the protective effect of aronia supplementation was diminished at this stage, suggesting a preventive rather than therapeutic approach. If animals were euthanized at 5 weeks it is not clear how authors discuss the effect of aronia at week 7

Response #13:  This part was to clarify the extent of inflammation observed in our prior study that evaluated a 7 week timepoint. We have revised this for clarity at LN 361-362:

We previously showed that at ….

Round  2

Reviewer 1 Report

While several points have been addressed in the revised version, the interpretation of the aronia-effect is overdrawn: Aronia berries do not inhibit instead they ameliorate inflammation. Inhibition is a complete block of colitis which is not depicted in Fig 1a. Please change the title accordingly.

Aronia supplementation improved essential immunological/inflammatory parameters specifically in the colon: weight lenght (Fig1B), reduced FDC uptake, lower inflammatory IL-6, TNF-a and IFN-g cytokine (Fig 2), reduced Malondialdehyde and enhanced antioxidant enzymes.

Thus, it is confusing that aronia does not improve the inflammation score of the colon and reduction of mitochondrial stress is only seen in the spleen (Fig 3a).

If colonic inflammation  is improved, the effect can not be projected on the spleen but takes directly place (by some or other mechanisms) in the target tissue. The argument that the spleen is full of immunocytes is not valid, as also the colon tissue is filled up with diverse immune and epithelial cells. 

Colonic epithelial cells might be a target of aronia. Thus the MitoP/MitoB ratio in colonic epithelial cells has to be checked, as these cells are also the main producers of IL-6 and TNF-apha in the colon.

The result section (line 242- 258) has to be revised, as the "colitic weight loss" is not disconnected from colonic inflammation.....this might be explained by a different behaviour of epithelial cells. 

 Author Response

RESPONSE TO PEER REVIEW

We sincerely appreciate the constructive comments provided by the peer reviewers and editor. The following is our reply and explanation of changes made to the manuscript considering these comments. Changes to the text are marked in red.

Comment #1: While several points have been addressed in the revised version, the interpretation of the aronia-effect is overdrawn: Aronia berries do not inhibit instead they ameliorate inflammation. Inhibition is a complete block of colitis which is not depicted in Fig 1a. Please change the title accordingly. 

Response #1:  We changed “inhibits” to “mitigates” in the paper and Fig. 1 titles. We do not mean to communicate a complete inhibition of inflammation. We believe “mitigates” is the most appropriate synonym for “partial inhibition” here.

Comment #2: Aronia supplementation improved essential immunological/inflammatory parameters specifically in the colon: weight length (Fig1B), reduced FDC uptake, lower inflammatory IL-6, TNF-a and IFN-g cytokine (Fig 2), reduced Malondialdehyde and enhanced antioxidant enzymes.

Thus, it is confusing that aronia does not improve the inflammation score of the colon and reduction of mitochondrial stress is only seen in the spleen (Fig 3a).

If colonic inflammation is improved, the effect cannot be projected on the spleen but takes directly place (by some or other mechanisms) in the target tissue. The argument that the spleen is full of immunocytes is not valid, as also the colon tissue is filled up with diverse immune and epithelial cells. 

Colonic epithelial cells might be a target of aronia. Thus the MitoP/MitoB ratio in colonic epithelial cells has to be checked, as these cells are also the main producers of IL-6 and TNF-alpha in the colon.  

Response #2:  

Regarding the spleen, we did observe increased inflammation in the spleen during colitis (Fig. 1c) that was most likely due to pro-inflammatory lymphocytes accumulation after T cell transfer. The spleen is the largest secondary lymphoid organ and mainly consists of resident lymphocytes and phagocytes [1]. The spleen filters blood-borne antigens and participates in the immune response to blood-borne bacterial, viral, and fungal infections. In addition, diseases can recruit additional immune cells to the spleen and elicit an immune response in spleen. For example, spleen enlargement has been observed upon adoptive transfer induced colitis in mice [2]. Therefore, the spleen participates in the colitic immune response and is affected by intestinal inflammation as demonstrated in Fig. 1c.

We agree that the colon tissue contains diverse immune and epithelial cells. The colon tissue used in the present study consists of epithelial cells, muscle cells, fibroblasts and collagen fibers. As a result, we cannot use our data to suggest that MitoP/MitoB ratios were altered in specific cells in the colon tissue. We agree with endothelial cells would also produce proinflammatory cytokines in the colon. The endothelial cells recruit professional immunocytes during inflammation, as specifically blocking adhesion molecules inhibits leukocyte-endothelial cell interaction and attenuates inflammation [3]. We agree with that colonic epithelial cells might be a target of aronia. However, it is not technically feasible to determine the MitoP/MitoB ratio in colonic epithelial cells (or colonic immunocytes), because the isolation of epithelial cells is a time-consuming process and involves enzymatic hydrolysis, repeated washing, which would affect the ratio and render questionable results.

We amended our discussion per these points at LN 414-424:

The different effects of aronia supplementation on mitochondrial H2O2 production in the spleen and colon during colitis could be explained by differences in cell populations between these tissues. We cannot exclude the possibility that colonic epithelial cells or immunocytes have altered mitochondrial H2O2 production after aronia consumptions, since we evaluated MitoP/MitoB ratios in whole colon tissue. Indeed, anthocyanin and other berry polyphenols can have direct effects on epithelial cells, which include inhibition of NF-κB, inhibiting cytokine production, and improving barrier function [45]. Isolation of specific cell types from the colon and other tissues requires additional time and processing, which may alter transient MitoP/MitoB distribution in viable cells. Thus, other experimental approaches are needed to define the role of mitochondrial H2O2 in specific cell types during colitis.

Comment #3: The result section (line 242- 258) has to be revised, as the "colitic weight loss" is not disconnected from colonic inflammation.....this might be explained by a different behavior of epithelial cells.

Response #3:  Per our response above, we did not collect data specifically on epithelial cell function in the small intestine or colon, but agree that aronia consumption could affect epithelial cells. We agree with that the colitic weight loss is associated with colitis, as we discussed at LN 358-361:

Loss of body weight and other symptoms of colitis (i.e. diarrhea or lose stool, reduced physical activity) result from systemic production of effector T cells (Th17 and Th1) after the T cell transfer and the interaction between effector T cells and gut microbiota.

We also added at LN 362: Impaired absorption of nutrients during colitis can be explained by changes in enterocyte functionality in the small intestine [35].

1.            Bronte, V.; Pittet, M.J. The spleen in local and systemic regulation of immunity. Immunity 2013, 39, 806-818.

2.            Capitan-Canadas, F.; Ocon, B.; Aranda, C.J.; Anzola, A.; Suarez, M.D.; Zarzuelo, A.; de Medina, F.S.; Martinez-Augustin, O. Fructooligosaccharides exert intestinal anti-inflammatory activity in the cd4+cd62l+t cell transfer model of colitis in c57bl/6j mice. Eur. J. Nutr. 2016, 55, 1445-1454.

3.            Panes, J.; Granger, D.N. Leukocyte-endothelial cell interactions: Molecular mechanisms and implications in gastrointestinal disease. Gastroenterology 1998, 114, 1066-1090.